# STRUCTURAL PRIVACY IN GRAPHS

## ABSTRACT

Graph Neural Networks (GNNs) gained popularity to address the tasks over the graph-structured data that best represent many real-world systems. The privacy of the participants of these systems is at risk if the GNNs are not carefully designed. Existing works in privacy-preserving GNNs primarily ensure the privacy of features and labels of a node. To ensure complete privacy related to graph data, its structure also needs to be privatized. We provide a method SPGraph to privatize the graph structure by adding noise to the neighborhood data of the node. Our method addresses two challenges in introducing structural privacy in graphs. Applying randomization on the set of actual neighbors to introduce noise leads to a reduction in the degree of nodes, which is undesirable. To overcome this first challenge, we introduce $\lambda$-selector that samples nodes to be added to the set of neighbors. The second challenge is to denoise the neighborhood so that the noise added in the neighborhood does not significantly impact the accuracy. In this view, we use the $p$-hop neighborhood to compensate for the loss of actual neighbors in the randomization.

We continue to use the node and label privacy as implemented in the previous methods for privacy in GNNs. We conduct extensive experiments over real-world datasets to show the impact of perturbation on the graph structure.

## 1 INTRODUCTION

Real-world systems such as social networks, citation networks, and molecular networks are popularly modeled as graphs. A graph richly represents such systems as it considers all the entities in the system as well as relationships between the entities. Graph Neural Networks (GNNs) are popularly used to tackle the tasks, such as node classification, graph classification, and link prediction are addressed using GNNs. The primary goal of GNNs is the aggregation of structural as well as feature information in an efficient manner. At the same time, it addresses the tasks related to the systems represented as graphs.

***Problem and Motivation.*** The problem we address in this work is ensuring the users' data privacy in critical systems that can be represented using graphs. The term *data privacy* in the case of graphs signifies privacy related to the structure of the graph and the features of the nodes in the graph. We look at the privacy in GNNs for node-level tasks. Most of the previous works in the area of privacy in GNN have ensured the privacy of the features and labels of each of the nodes. It assumes that the server knows the connectivity information; hence in the previous works, the term data privacy means the privacy of node features and node labels. In this work, we consider the privacy of structural information, typically given by the edges in the graphs, along with the feature and label information of the nodes.

***Challenges.*** To ensure complete privacy, that is, the privacy of nodes, edges and labels in a graph, the following challenges need to be addressed: 1. Edge privacy or the privacy of the graph structure is to be taken care of to avoid the information from being compromised. The existing methods Sajadmanesh & Gatica-Perez (2021) perturb only the node features and the node labels, however as the graph structure remains publicly available, we need to consider that as private information as well. 2. To preserve structural privacy, there has to be a mechanism to perturb the edges in the graph. Deciding the amount of noise to be added to the edge data and the mechanism to add noise is one of the challenges. We also need a mechanism to correct the added noise so as to balance the accuracy of the predictions. 3. Determining the amount of structure perturbation and node and label perturbation to strike the right balance between privacy and utility in the graph data.

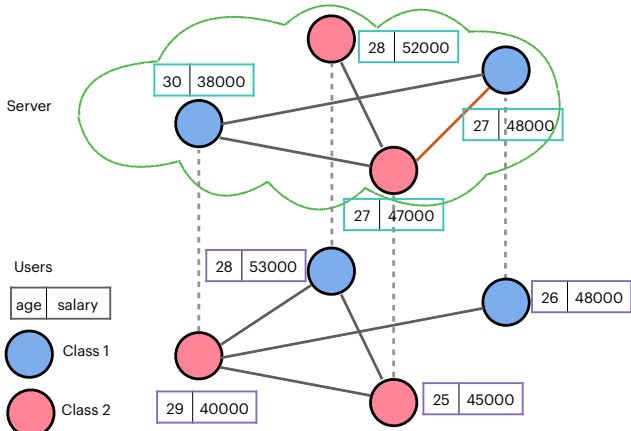

Figure 1: A network of users where users want to keep their sensitive data such as age and salary private. This information is privatized by perturbing the node feature and label information. The change is labels is shown with a change in the node color. We address the problem of structure perturbation in such a network before passing on all the perturbed information to the server to further process the data by applying GNN to answer the questions on the graph.

***Contributions.*** We propose completely locally private graph neural networks. Complete local privacy is obtained by privatizing the edge information in addition to node and label information. Our method SPGRAPH privatizes the structure of the nodes in the graph data. We provide experimental evidence of the performance of the model by varying the parameters that control the noise added to the edges for perturbations where we perturb the edges according to the structure privatization approach.

***Paper Organization.*** This paper is organized as follows. Introduction, motivation, challenges, and contribution of our work are described in section 1. In section 2, we discuss the works related to the topic. Section 3 gives the preliminaries with problem definition and background. Our proposed method is described in section 4. The experimental setup and the results are discussed in section 5 and section 6 concludes the paper.

## 2 RELATED WORKS

With different possible attacks on Graph Neural Networks, the privacy of the data involved can be compromised. He et al. (2021) introduced seven different link stealing attacks on Graphs. The adversary has black-box access to Graph Neural Networks, and it can infer whether or not any two nodes used in the training of the GNN have a link between them. Wu et al. (2021b) introduced a similar attack called LinkTeller attack that concerns the privacy of the edges in the graph. In a setting where the node features and the adjacency information are with different parties, the party with the adjacency matrix trains the GNN upon receiving the node features from the other party and provides back the inference API. The party holding the node features provides the test node features and also can query API for predictions related to test nodes. The LinkTeller attack tries to infer the links present between nodes, based on the queries. The other works Olatunji et al. (2021), Zügner et al. (2020), Duddu et al. (2020) discuss the attacks possible on GNNs, such as membership inference attack, graph reconstruction attack, and attribute inference attack.

A federated framework for privacy-preserving GNNs for recommendation systems is presented in Wu et al. (2021a). The GNNs are trained locally at the users' end and they upload the local gradients to a server for aggregation. To enhance the privacy of users by protecting user-item interaction, local differential privacy techniques are applied to the locally computed gradients. Zhou et al. (2020) is another work involving federated graph neural networks for privacy-preserving classification tasks, the features, and the edges are split among the users, while all the users have the access to the same set of nodes.

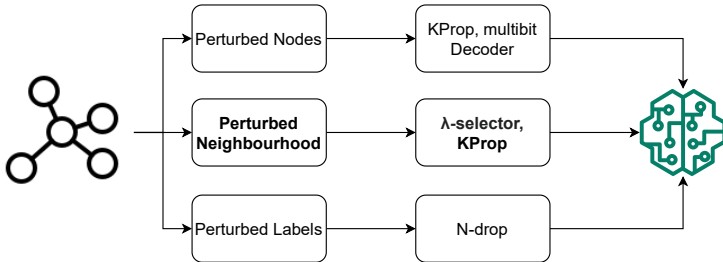

Figure 2: To ensure complete privacy, in addition to node and feature perturbation we perturb the neighborhood using SPGRAPH that uses $\lambda$-selector and KProp.

To get the differentially private node embeddings, DPNE Xu et al. (2018) applies objective perturbation on the objective function of matrix factorization. Sajadmanesh & Gatica-Perez (2021) proposes a privacy-preserving GNN learning algorithm for the privacy of nodes where the node features and their labels are assumed to be private while the structure of the graph is not private.

## 3 PRELIMINARIES

### 3.1 PROBLEM DEFINITION

**Definition 3.1 (Graph)** *A graph $G$ is defined as $G = (V, E, X, Y)$ where, $V = V_L \cup V_U$ is a set union of labeled nodes $V_L$ and unlabeled nodes $V_U$ and $|V| = n$. $E$ is a set of edges and $|E| = m$, $X$ is the feature matrix corresponding to the graph where $X \in \mathbb{R}^{n \times d}$ and $d$ is the size of feature vector corresponding to each of the nodes in the graph. $Y \in \{0, 1\}^{n \times c}$, where $c$ is the total number of classes nodes belong to, is the matrix of one-hot encodings of the labels corresponding to labeled nodes and it is a vector of all zeros for the unlabeled nodes in the graph. The neighborhood of a node $v$ is given as $\mathcal{N}(v)$ that contains nodes $u$ such that $(u, v) \in E$.*

Consider a client-server system where the clients are the users represented by nodes and the server is where the Graph Neural Network runs and predicts the labels of unlabeled nodes. The nodes are aware of their feature vectors and their neighbors. The server is aware only of the vertex set $V$. The GNN needs to be trained for the given task on the server side. The problem we try to address is how to train the GNN on the server side while preserving neighborhood privacy in addition to preserving the privacy of node features and labels.

### 3.2 BACKGROUND

#### 3.2.1 GRAPH NEURAL NETWORK

Graph neural networks are typically used to answer questions based on some graph data. The tasks such as node classification, link prediction, and graph classification can be addressed using GNNs. A graph $G = (V, E)$ with a matrix feature vectors $X \in \mathbb{R}^{n \times d}$ where $n = |V|$ and $d$ is the size of the feature vector $x_v$ of node $v \in V$. We represent the intermediate embedding of node $v$ at $l$th layer as $h_v^l$. The final representation of each node is then passed to the downstream deep network so as to make the desirable predictions related to the task. To get the representation of each node in the graph, a GNN has two primary steps at each layer for every node: 1. Aggregation of node's neighborhood and 2. Updating the node's embedding. These steps are applied as many times as the number of layers in the GNN.

$$h_{\mathcal{N}_v}^l = \text{AGGREGATE}\left(\{h_u^{l-1} : u \in \mathcal{N}(v)\}\right) \tag{1}$$

$$h_v^l = \text{UPDATE}\left(\{h_v^{l-1}, h'_{\mathcal{N}_v}\}\right) \tag{2}$$

In equation 1 above, AGGREGATE is a differentiable, permutation invariant function that aggregates the feature vectors from layer $l - 1$ of the nodes in neghiborhood $\mathcal{N}(v)$ of the node $v$. The neighborhood is defined as $\mathcal{N}(v) = \{u : (u, v) \in E\}$. In equation 2, UPDATE is a differentiable function.

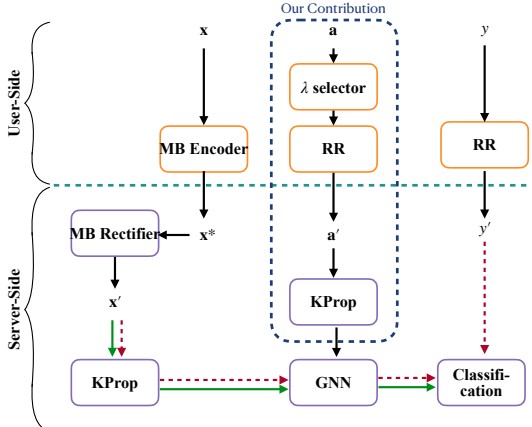

Figure 3: Overview of SPGraph algorithm with edge privacy in addition to the node features and label privacy. The nodes' structure is perturbed using the $\lambda$ selector and then applying randomized response on the set of neighborhoods.

### 3.2.2 DIFFERENTIAL PRIVACY

Local differential privacy (LDP) enables users to share noisy private data with the untrusted aggregator instead of their true private data while maintaining considerable accuracy in group queries. It has been used in the last decade for collecting sensitive private data and answering group queries such as statistical mean and count. Companies like Apple Cormode et al. (2018), Google Úlfar Erlingsson et al. (2014), and Microsoft Ding et al. (2017) have already started including LDP in their products.

**Definition 3.2 (Local Differential Privacy)** *An algorithm $\mathcal{A}$ satisfies $\epsilon$-local differential privacy ($\epsilon$-LDP), if and only if for any input $x$ and $x'$, we have*

$$\forall y \in Range(\mathcal{A}) : Pr[\mathcal{A}(x) = y] \leq e^\epsilon Pr[\mathcal{A}(x') = y] \tag{3}$$

*where Range($\mathcal{A}$) is the set of all possible outputs of the algorithm $\mathcal{A}$ and $\epsilon \geq 0$.*

## 4 PROPOSED METHOD

This section describes our method to introduce structural privacy in Graph Neural Networks. For a particular node $v$, we add the noise to the set of its neighbors. This will ensure that the actual neighborhood information is not passed to the server. A neighbor is either dropped from the set of neighbors on the application of the randomization mechanism. It will keep or remove the neighbors with some associated probability. Figure 4.1 shows the algorithm, which we describe step-by-step here.

### 4.1 PRIVACY IN GRAPH STRUCTURE

To preserve privacy in the graph's structure, we perturb the connectivity information of the graph. This is done by introducing noise in the neighborhood of the nodes. We propose the methodology that we call $\lambda - selector$ to induce noise and is given in algorithm 1.

The noise is added as shown in the algorithm 1. This, however, needs to consider the following things.

### 4.1.1 CHOOSING THE VALUE OF $\lambda$

Our method considers the set $\mathcal{N}(v)$ of actual neighborhood nodes on the node $v$. We cannot randomize this set of nodes, as the randomization will only lead to removing some of the existing neighbors. It will result in a reduction in the degree of the node and also a reduction in the average

---

**Algorithm 1** $\lambda$ Selector Algorithm

---

**Input:** $v$: A node in the vertex set $V$, $\sqsubseteq$: The set of neighboring nodes of node $v$, $\lambda$: Number of extra nodes to be added to $\mathbf{a}$, $V_{C_v}$: The set of nodes in the cluster containing node $v$

**Output:** $\mathcal{N}'(v)$: A noisy set of neighbors of node $v$

1: $\mathbb{M} \leftarrow$ Sample a set of nodes the set $V_{C_v} \backslash \mathbb{A}_v$ , uniformly at random without replacement
2: $\mathbb{A}_v^* \leftarrow$ apply randomization mechanism on the set $\mathbb{A}_v \cup \mathbb{M}$
3: **return** set $\mathcal{N}'(v)$ as the noisy neighborhood set

---

degree of the nodes in the graph. With backtracking, it is still possible that the actual neighborhood information is compromised. Additionally, there is no noise added to the neighborhood information when randomization is applied only on the original neighborhood. For this reason, we consider $\lambda$ more nodes in addition to $\mathbb{A}_v$ for applying randomization. The hyperparameter $\lambda$ is to specify how many more nodes to consider in addition to the actual neighborhood. A randomization mechanism is applied to the set of nodes with the actual neighborhood nodes plus $\lambda$ sampled nodes.

When the value of $\lambda$ is too high, the privacy budget gets reduced significantly per neighborhood node, resulting in the addition of less noise. On the other hand, when the value of $\lambda$ is too less, though it will add noise to the actual neighborhood, it mostly results in the subsets of the neighborhood or a very less number of additional nodes depending on the value of $\lambda$. Both of these scenarios are not desirable. Hence the choice of the value of $\lambda$ is important.

### 4.1.2 Set to sample $\lambda$ nodes from

The choice of the value of $\lambda$, as well as the set from which the $\lambda$ many nodes are chosen, is crucial. We apply randomization over all nodes in the graphs if we select $\lambda = n - d_v$, the randomization will be applied to all the nodes. The privacy budget $\epsilon_e$ for edge perturbation gets distributed over $n$ nodes. Hence, the privacy budget becomes $\epsilon_e/n$ per neighbor. The noise is proportional to the privacy budget, due to which $\epsilon_e/n$ privacy budget per neighbor does not add a significant amount of noise in the neighborhood, as the privacy budget gets divied across all the $n$ nodes in the graph, where $n$ is quite large.

As we understand that in a graph, similar nodes are connected, choosing $\lambda$ randomly from the set of all nodes except the actual neighborhood nodes may lead to sampling nodes that are many hops away from the node in consideration, $v$. These nodes far away from $v$ may not be similar to it. Given the neighborhood aggregation in a GNN, the neighbors contribute significantly to the node's embedding. If irrelevant nodes are added to the neighborhood, they would highly affect the final embedding of the node. This is unlike the desirable case where only the relevant neighbors impact the node's final embedding. Adding distant nodes to the neighborhood would result in a drop in accuracy.

To overcome this challenge, we propose considering the nodes in the cluster to which the node $v$ belongs using a differentially private clustering algorithm to sample $\lambda$ nodes. Differentially private clustering algorithms like Stemmer (2020); Bun et al. (2021); Mülle et al. (2015) can be used to get clusters from which we can sample $\lambda$ nodes. This can avoid adding distant nodes as neighbors and keep the perturbed edge list confined to the local cluster. Other methods of differentially private sharing of neighbors upon a query to get a noisy neighborhood may be used as well.

### 4.1.3 Addition of noise by randomization

Once we have the set of nodes sampled from the clusters and the actual neighbors, we can apply a randomization mechanism to add noise to the data. We use the randomized response as the randomization mechanism. For each user, adjacency vector $a$, randomized response algorithm creates a perturbed edge $a_i^{'}$ for every edge $a_i$.

This vector $a_i^{'}$ is the perturbed neighborhood the user can share with the server. This ensures the privacy of the graph's structure when every node is sharing the perturbed neighborhood.

**Lemma 4.1** *Edge perturbation through randomized response satisfies $\epsilon$-local differential privacy Dwork et al. (2014).*

### 4.1.4 Removal of added noise

Adding noise to the neighborhood means that some nodes are not connected to the node in the original graph, but the edge exists in the perturbed neighborhood. This will lead to the node being connected to the non-neighboring nodes, which will contribute to the nodes' final embeddings and affect the predictions done by the GNN. We leverage the KProp algorithm Sajadmanesh & Gatica-Perez (2021) to cancel this noise. It aggregates the representations of the neighbors that are up to $k$ hops away. Initially in section 4.1.2, we carefully choose the set from which we sample $\lambda$. As these are nodes in the vicinity of the node in question, we do not deviate much from the embedding that the node would have produced with the original neighborhood. The node embedding is updated as shown in the equation 4 in parallel for $k$ times for all the nodes in the graph.

$$h_{\mathcal{N}'(v)}^k = \text{AGGREGATE}\left(\{x_u^{k-1}, \forall u \in \mathcal{N}'(v) - \{v\}\}\right) \tag{4}$$

$$\hat{h}_v = \text{UPDATE}(h_{\mathcal{N}'(v)}^K) \tag{5}$$

where $\mathcal{N}'(v)$ is the perturbed neighborhood of node $v$.

## 4.2 Privacy in Node Features

Node features are privatized using *multi-bit encoder*, which is built upon 1-bit mechanism, on the user-side. The feature vector corresponding to a node is requested by the server, the multi-bit mechanism is applied on it to get the encoded feature vector which is sent to the server. This gives a biased output which is rectified on server-side using a *multi-bit rectifier*. The differential privacy budget for node feature perturbation is $\epsilon_x$. The estimation error is inversely proportional to the number of neighbors of a node, but in real graphs, the size of the neighborhood is usually small. To overcome this issue KProp layer is used as a first GNN layer in order to denoise the input to the GNN. It considers the $k$-hop neighborhood of a node for denoising the perturned node features at server-side.

## 4.3 Privacy in Node Labels

Each node participating in training has to perturb the labels as they are considered private. This can be done using an LDP mechanism, which in this case is Randomized Response. The perturbed labels are sent to the server where KProp is applied on node labels. GCN aggregator is used as the aggregate function in KProp as it leads to lower estimation error. The differential privacy budget for node feature perturbation is $\epsilon_y$. The label denoising with propagation (Drop) Sajadmanesh & Gatica-Perez (2021) is used for training using perturbed node features and node labels.

## 4.4 Privacy analysis of SPGraph

For each user, adjacency vector $a$, randomized response algorithm creates a perturbed edge $a_i'$ for every edge $a_i$. Here $a_i$ could be 0 or 1. Randomized response algorithm is defined as below:

$$a_i' = \begin{cases} 1, & \text{with probability } \frac{1}{2}f \\ 0, & \text{with probability } \frac{1}{2}f \\ a_i, & \text{with probability } 1 - f \end{cases} \tag{6}$$

Relevant probabilities are:

$$P(a_i' = 1 | a_i = 1) = \frac{1}{2}f + 1 - f = 1 - \frac{1}{2}f \tag{7}$$

$$P(a_i' = 1 | a_i = 0) = \frac{1}{2}f \tag{8}$$

In the algorithm, $\lambda$ *selector*, the algorithm keeps the $d - \lambda$ neighborhood intact and only applies randomized response to the $\lambda$ percentage of nodes. Without loss of generality, let the adjacency vector to be perturbed be represented by i.e. $a^* = \{b_1 = 1, ..., b_i = 1, b_{i+1} = 0, ..., b_n = 0\}$.

Let the $RR$ represent the ratio between adjacent neighborhood $a'$ and $a$, which differ by only one edge. Let the $a'$ differ at the $i$th position as the perturbed neighborhood only differs by one row. Given the independence of each edge, we only need to consider the edge that changed.

$$RR = \frac{P(\mathcal{A}(a) = y)}{P(\mathcal{A}(a') = y)} \leq max \frac{P(a_i' = 1 | a_i = 1)}{P(a_i' = 1 | a_i = 0)} = \frac{1 - \frac{1}{2}f}{\frac{1}{2}f} \tag{9}$$

$$\epsilon' = ln(RR) = ln(\frac{1 - \frac{1}{2}f}{\frac{1}{2}f}) \tag{10}$$

We combine perturbations in node features, labels, and neighborhoods with privacy budgets $\epsilon_x$, $\epsilon_y$, and $\epsilon_e$ respectively in SPGRAPH.

**Theorem 4.2** *Algorithm* SPGRAPH *satisfies* $\epsilon_x + \epsilon_y + \epsilon_e$-*local differential privacy.*

Theorem 4.2 follows from the post-processing composition theorem Sengupta et al. (2021). We process only the output of the LDP mechanisms on features, labels, and edges of the node. LDP mechanism is applied only once on the nodes, and hence any post-processing does not affect differential privacy Dwork et al. (2014).

### 4.4.1 SPGGRAPH'S IMPACT ON EMBEDDING

To a node $v$'s embedding, on average, the contribution of each of its neighbors can be thought of as $\frac{h_v}{|\mathcal{N}'(v)|}$. Similarly, a node that is $p$ hop away from the node $v$ will contribute $\frac{1}{\deg(v)^p} h_v$ to the node $v$'s embedding. Considering this, after the perturbation, the contribution to the new embedding from the $\lambda$ sampled nodes that belong to the cluster with radius $p$ is given as $\lambda \frac{1}{\deg(v)^p} h_v$. The ratio of the embeddings of node $v$ before and after perturbation is $\frac{h_v}{\hat{h}_v} = \frac{h_v}{\lambda \frac{1}{\deg(v)^p} h_v}$ which comes to $\frac{\deg(v)^p}{\lambda}$. It should be noted that this analysis is based on the assumption that the neighborhood nodes contribute equally to the generated embedding of the node.

## 5 EXPERIMENTS

### DATASET DESCRIPTION

We perform the experiments on four popular datasets, namely Cora and Pubmed Yang et al. (2016), Facebook Rozemberczki et al. (2021) and LastFM Rozemberczki & Sarkar (2020). The dataset statistics are as given in the table 1.

1. *Cora and Pubmed*Yang et al. (2016): These are citation networks, where each node represent a document and if a document $i$ cites document $j$, then $(i, j) \in E$ and $(j, i) \in E$. The feature vector corresponding to a node is the bag-of-words representation of the document. The labels are the categories the document belong to.

2. *Facebook*Rozemberczki et al. (2021): Nodes in this dataset are verified Facebook pages and edges are the mutual likes between them. The classification is on the site caterogy and the feature vector are extracted from the site desciption.

3. *LastFM*Rozemberczki & Sarkar (2020): Nodes represent the users of radio streaming service LastFM, while the edges represent the friendships amoung the users. The nodes' features are constituted based on the artists a user likes. The classification task is to predict the home country of the user. The current usage of the dataset is only limited to top 10 countries.

### 5.1 EXPERIMENTAL SETUP

We conduct experiments on the all four datasets to study the effect of the addition of noise in the edge data and the effect of varying privacy budgets for edges in the graph. The experimental setup for privacy of node features and node labels remains similar to that in LPGNN Sajadmanesh & Gatica-Perez (2021).

The nodes in all the datasets are split in the ratio of 50:25:25 in training, validation, and test sets. The node features are normalized between [0, 1]. The privacy is applied on the graph neural network

Table 1: The statistics of the datasets

| Dataset | Classes | Nodes | Edges | Features | Average Degree |
|---------|---------|-------|-------|----------|----------------|
| Cora | 7 | 2708 | 5278 | 1433 | 3.90 |
| Pubmed | 3 | 19717 | 44324 | 500 | 4.50 |
| Facebook | 4 | 22470 | 170912 | 4714 | 15.21 |
| LastFM | 10 | 7083 | 25814 | 7842 | 7.29 |

models namely GCN Kipf & Welling (2016), GAT Veličković et al. (2017), and GraphSAGE Hamilton et al. (2017). Feature perturbation and edge perturbation are applied during training, validation, and testing, and the label perturbation is applied during the training and validation only. The privacy budgets for edge perturbation $\epsilon_e$ are taken from the set $\{0.01, 1, 2, 8, \infty\}$. We vary the amount of noise (in percentage) $\lambda$ added to the edges for perturbing the neighborhood of the nodes. We select three values $\deg, \deg/2$, and $\deg/4$ for $\lambda$ based on each dataset's average degree $\deg$. The set of lambda values for Cora is $\{4, 2, 1\}$, for Pubmed, it is $\{5, 3, 1\}$, for Facebook, it is $\{15, 8, 4\}$ and for LastFM, it is $\{7, 4, 2\}$. The other parameters for training using the Drop are the same as that of LPGNN Sajadmanesh & Gatica-Perez (2021).

For experimental purposes, in place of private clustering algorithms, we consider the $p$-hop neighborhood of the node for simplicity. In this place, any private clustering algorithm can be used. We experiment with values $k = 1, 2, 3$, and $\infty$ with $k = \infty$ considering all the other nodes in the graph. The sampling of $\lambda$ neighborhood nodes is done from these clusters.

We use Adam optimizer to train the model, and based on validation loss, we pick the best model for testing. Our code will be available on Github.

## 5.2 EXPERIMENTAL RESULTS

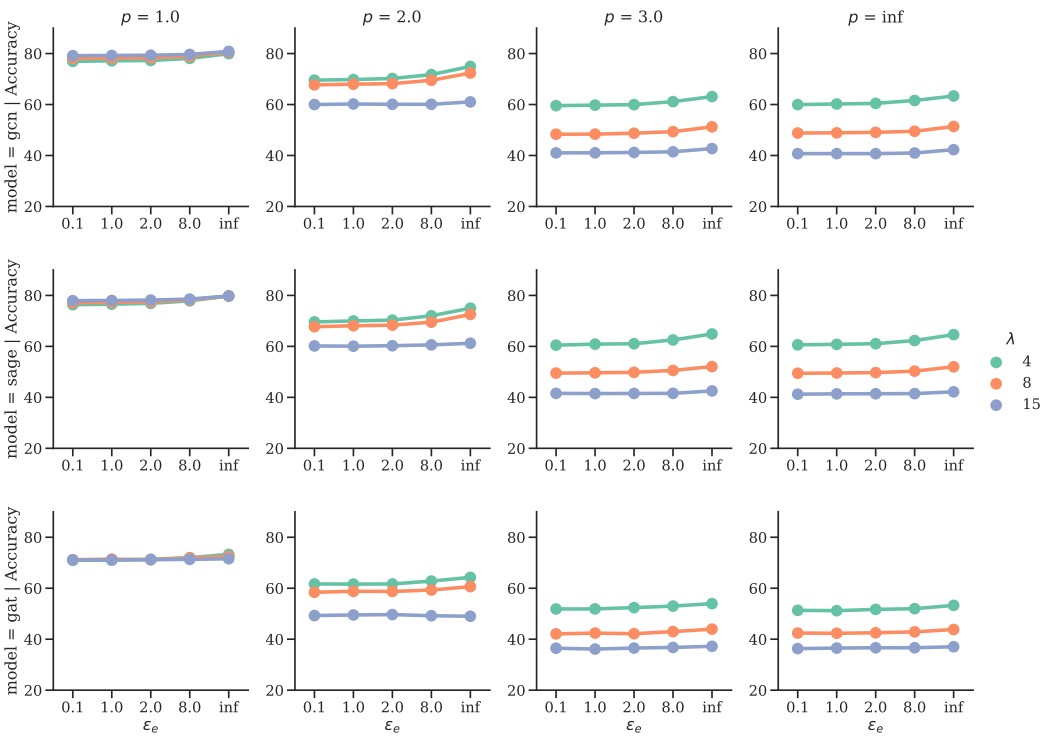

Figure 4: Accuracy for by SPGRAPH with different values of $\lambda$, $p$, and $\epsilon_e$ for Facebook dataset.

We vary the added noise $\lambda$, the radius $p$ of the clusters, and the privacy budget $\epsilon_e$ for the edge perturbation. The privacy-accuracy trade-off is well observed in the results. The hyperparameter $\lambda$ decides the amount of neighborhood that will be perturbed. In figure 4, we can see the lambda values' effect on the LastFM dataset. We see a drop in accuracy as we increase the value of $\lambda$. This is a result of the fact that the $\lambda$ contributes to the privacy of the neighborhood data. As $\lambda$ increases, we are sampling more nodes from the set of nodes that are not the actual neighbors of a node. This results in the contribution of more noisy data to the node's embedding. Depending on how much noise is added, the prediction accuracy falls.

The amount of perturbation in the selected edges is given by the privacy budget $\epsilon_e$ for edge perturbation. We experiment with the different values of privacy budget $\epsilon_e$ for edge perturbation. When the privacy budget is less, we expect the model to be more private and hence also to be less accurate. We see in figure 4 the effect of the privacy budget on the accuracy of different GNN models we use for the LastFM dataset. As the privacy budget increases, the amount of noise added through randomization decreases, leading to better accuracy.

To sample the $\lambda$ from, we carefully select a set as discussed in section 4.1.2. As for our experimental purposes, we use the $p$-hop neighborhood, which can also be seen as the radius of the private clusters. We see the effect of different values of $p$. We see our intuition behind our approach working from our experiments, which leads to the conclusion that the accuracy is better when the value of $p$ is less. This is because the lower radius of the cluster means that the neighborhood is closer to the node. With an increase in the radius, we tend to sample from distant nodes that may not be related to the node we are interested in. The effect of value of $p$ is shown in figure 4.

We provide experimental results depicting the effects of the values of $\epsilon_e$, $\lambda$, and $p$ in the appendix for the rest of the datasets. In some datasets, such as Cora, that have a less average degree, we see that as the value of $\lambda$ increases, the accuracy also increases, which is not in line with our earlier claim. This can be seen in figures 5, 6, 7 This exception is seen because in the dataset with a smaller average degree, such as 4 in the case of Cora, dropping one actual neighborhood also contributes significantly to the embeddings. Similarly, adding the distant nodes would lead to a drop in accuracy.

We also learned from these results that the model accuracy is better when the samples for adding noise to the node's neighborhood are chosen from smaller clusters. As the size of the clusters increases, the accuracy drops. This is justified as the smaller cluster enables sampling of nodes similar to the node to which the noise is added. The addition of similar nodes does not affect the final embeddings much and hence gives better accuracy than the sampled from the clusters with larger sizes.

## 6 CONCLUSION

In this paper, we introduced an approach SPGRAPH to ensure the privacy of the graph structure. It solves an important issue of having a privacy-preserving mechanism that ensures that the entire graph data is privately shared with the server. We discuss the $\lambda$ selector that samples $\lambda$ nodes from a set. Adding $\lambda$ nodes is essential as that adds noise to the structural information. We then study the significance and selection of the value of $\lambda$ and the set from which it should be sampled. To compensate for the added noise in the neighborhood, we utilize the method KProp to denoise the neighborhood so that the added noise does not significantly hamper the final prediction's accuracy. We demonstrate that our method upholds the privacy-accuracy trade-off after extensively experimenting with various cases of perturbations and values of different parameters that control the addition of noise to the graph's structure.

To further improve privacy in overall graph data, we list a few directions for future work. Firstly a better sampling mechanism can be explored. Specifically, the set from which the noisy neighborhood is sampled can be improved to include distant neighbors that are similar to give better prediction accuracy. Secondly, the existing mechanism for node feature perturbation provided in the earlier works is inadequate as it tends to lose substantial feature information. Better methods that can retain most of the feature information can be explored. Additionally, tighter bounds on privacy could be provided. And finally, similar privacy-preserving mechanisms can be explored for the graphs that assume heterophily among the nodes.

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

## A    ADDITIONAL RESULTS

This section demonstrate the results in addition to the ones reported in section 5.2.

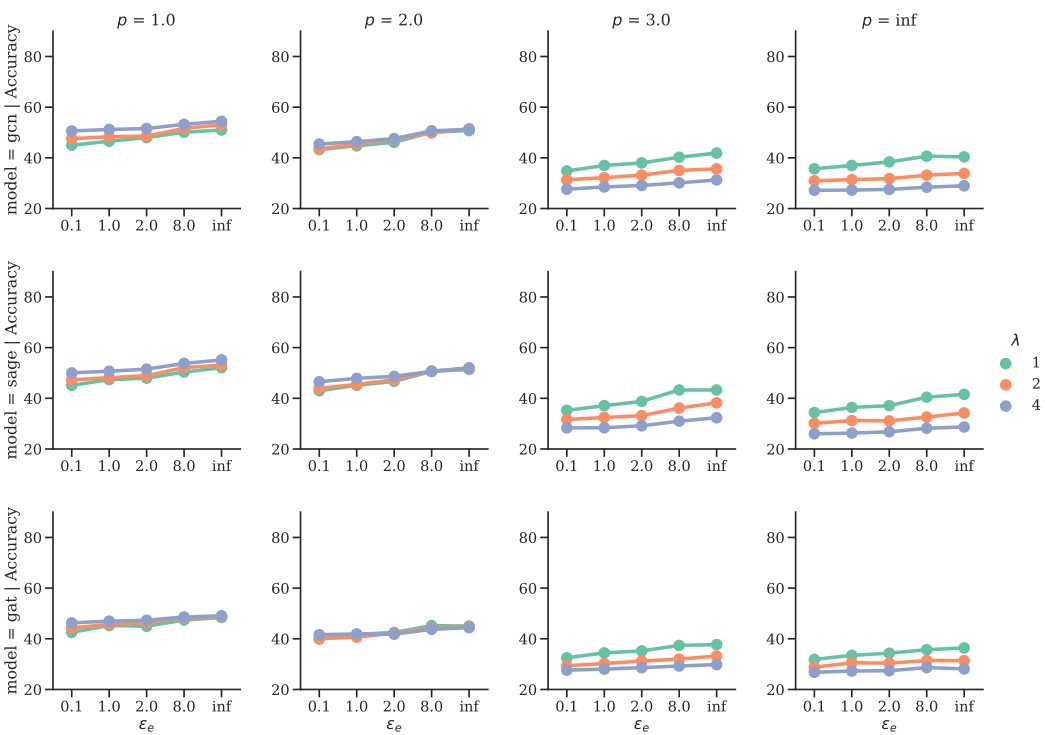

Figure 5: Accuracy for by SPGRAPH with different values of $\lambda$, $p$, and $\epsilon_e$ for Cora dataset.

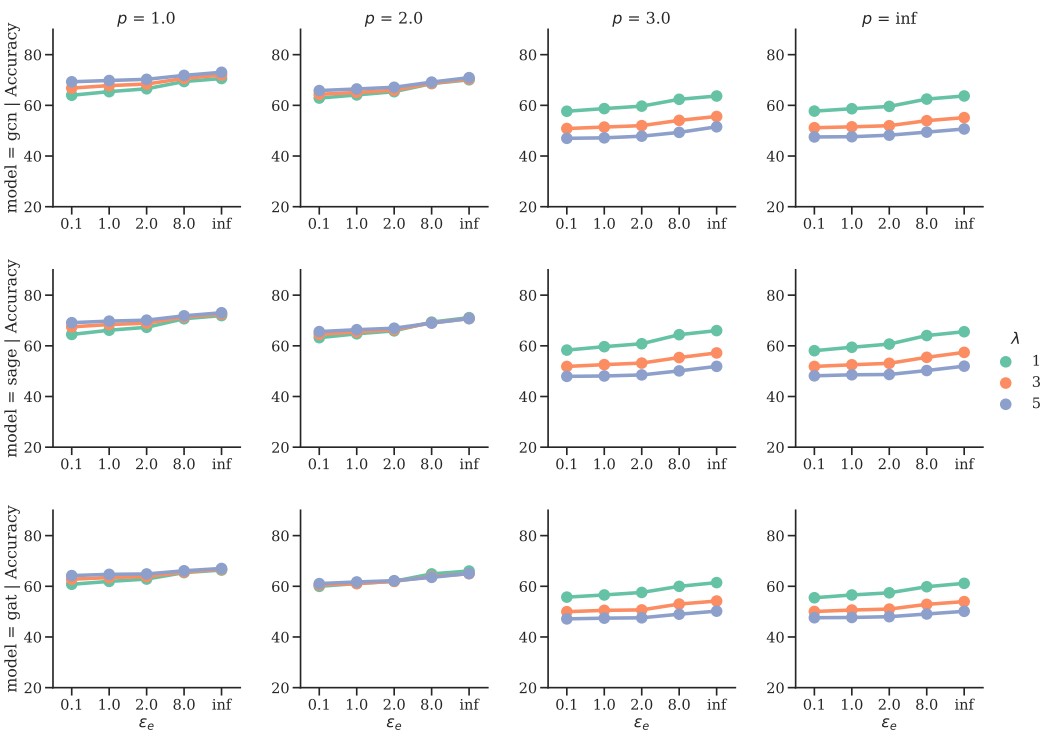

Figure 6: Accuracy for by SPGRAPH with different values of $\lambda$, $p$, and $\epsilon_e$ for Pubmed dataset.

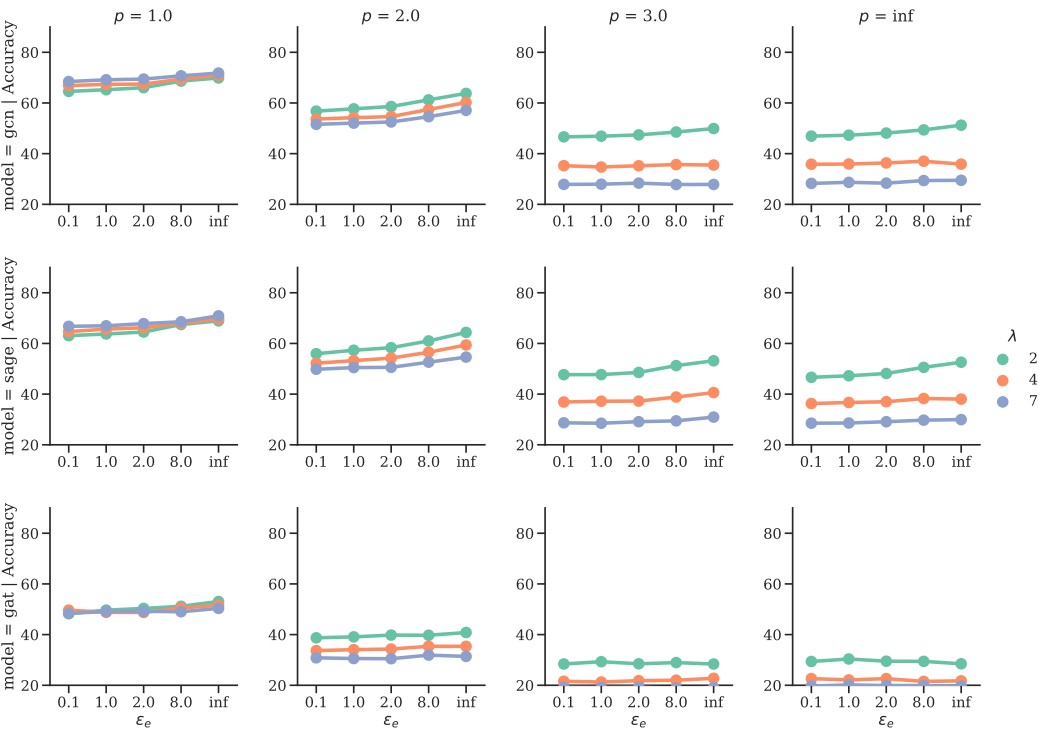

Figure 7: Accuracy for by SPGRAPH with different values of $\lambda$, $p$, and $\epsilon_e$ for LastFM dataset.

