# OpenReview forum: "Structural Privacy in Graphs"
_ICLR.cc/2023/Conference — Submitted to ICLR 2023_

### Official Review · Reviewer_nZ2k · 2022-10-23

**Confidence:** 4
**Correctness:** 1
**Technical Novelty And Significance:** 2
**Empirical Novelty And Significance:** 1
**Recommendation:** 3

**Clarity, Quality, Novelty And Reproducibility:**

The paper does not present results that could be reproduced on the privacy side of the accuracy-privacy tradeoff.

**Strength And Weaknesses:**

Stength:
Proposes a reasonable method for graph sharing by noise addition.

Weakness:
Does not offer intelligible and consequential privacy guarantees for sharing the whole graph structure as claimed.
No results on the privacy side of privacy-accuracy tradeoff.

**Summary Of The Paper:**

This paper proposes an method that aims to protect the privacy of sharing a graph structure.
A claim is made that an entire graph is to be privately shared, without a clear definition of how such whole-graph privacy would arise.
The type of noise addition following constitutes the addition of nodes sampled from a set by a local-differentially private clustering algorithm, while the added noise is compensated for by a Kprop method, derived from previous work, that aggregates the representatives of neighbor up to k hops away.
An experimental study claims to examine the tradeoff between privacy and accuracy.

**Summary Of The Review:**

This paper proposes an method that aims to protect the privacy of sharing a graph structure.
A claim is made that an entire graph is to be privately shared, without a clear definition of how such whole-graph privacy would arise.
The type of noise addition following constitutes the addition of nodes sampled from a set by a local-differentially private clustering algorithm, while the added noise is compensated for by a Kprop method, derived from previous work, that aggregates the representatives of neighbor up to k hops away.
An experimental study examine the tradeoff between privacy and accuracy.

However, it is not clear what privacy guarantees the proposed method offers in practice. The differential privacy guarantee, in a graph context, would imply that a difference of one graph element would not be noticeable to an adversary under noise addition. However, this guarantee says nothing about the ability to derive large-scale graph properties, which could threaten the privacy of individual indeed. Past research has shown [CIKM 2012] that most graph measures can be recovered under noise addition, and that it is hard to even obscure the length of a path in a graph structure. This paper appears to be incognizant of such concerns, as well as of the limitations of differential privacy when it comes to sharing correlated structural idea. Thus, it is not clear what consequences the proposal local differentially privacy algorithms have in effect. A discussion of these concerns is due.

---

### Official Review · Reviewer_Q4eW · 2022-10-24

**Confidence:** 5
**Clarity, Quality, Novelty And Reproducibility:** See strength and weaknesses.
**Correctness:** 2
**Technical Novelty And Significance:** 1
**Empirical Novelty And Significance:** 1
**Recommendation:** 3

**Strength And Weaknesses:**

Strength:

1) The problem of preserving the privacy of the graph structure is an interesting problem and may be practically useful.

2) According to the claim of this paper, this is the first work trying to investigate the structural privacy problem in GNNs.


Weaknesses:

1) The proposed method and analysis do not seem to be specific to GNNs. It is weird that one can claim the privacy of the GNN training algorithm without involving the GNN in the analysis.

2) The lemma and theorem are essentially stated without proof. Either they are incorrect or they are direct applications of existing results. In either case, there is little novel technical contribution.

3) The clarity can be improved. Below are some concrete cases that are unclear in writing.

- Section 4.1.3 states that "We use the randomized response as the randomization mechanism". The "randomized response" method should be explained with citations.
- Algorithm 1 states that randomization will be applied to edges to the neighborhood of each node plus lambda nodes. But section 4.1.3 states that randomization will be applied to every edge.
- Lambda is defined as an integer in Algorithm 1 indicating a number of nodes. In section 4.4, there is a phrase "lambda percentage of nodes", which is confusing.

4) Typos.

- Page 5. "... is too less, ..." -> is too small
- Page 5 and Algorithm 1. A_v not defined. There seems to be a change of notation from N(v) to A_v.




**Summary Of The Paper:**

This paper investigates the problem of preserving the privacy of edges in the training graph in GNNs. The authors propose to perturb the edges of each node to achieve local differential privacy.

**Summary Of The Review:**

While this paper investigates an interesting problem, the proposed method and analysis seem trivial or wrong.

---

### Official Review · Reviewer_nodK · 2022-11-01

**Confidence:** 4
**Correctness:** 3
**Technical Novelty And Significance:** 2
**Empirical Novelty And Significance:** 2
**Recommendation:** 1

**Clarity, Quality, Novelty And Reproducibility:**

Algorithm 1 (which if referred to in Sec 4,1) :
- the symbol \sqsubseteq is often used to express a relation, rarely to denote a set of neighboring nodes.  why dont you use the common notation N_v ?
- “… extra nodes to be added to a.” : the variable “a” is not defined yet
- V_{C_v} : why do you need a double subscript?  C_v on itself is not yet given a meaning.
- cluster containing v: where is this cluster defined?
- line 1: “ Sample a set of nodes the set” -> … from the set
- V_{C_v} \setminus \mathbb{A}_v : the variable \mathbb{A}_v is not yet defined
- line 2: “apply randomization mechanism on” : what randomization mechanism should be applied?
- line 3: “ return set N ′(v) as the noisy neighborhood set” : the variable N’(v) is not yet defined.

In conclusion, at the point alg 1 is referred to and in its current form, it is very hard to understand alg 1,  Similar concerns hold for other parts of the text.

**Details Of Ethics Concerns:**

—

**Strength And Weaknesses:**

The writing, the formalization and the English language used are rather poor, making it often hard or impossible to infer the meaning of the text.

In general, the paper may have interesting ideas, but it is hard to appreciate then given the current writing.  Also, it is unclear whether the proposed result is very deep, it may be a direct application of earlier work by Dwork (lemma 4,1).


**Summary Of The Paper:**

This paper proposes a strategy to add noise / privacy to the structure of a graph.

**Summary Of The Review:**

My main concern is that the paper is insufficiently clear.

---

### Decision · Program_Chairs · 2023-01-20

**Decision:**

Reject

**Justification For Why Not Higher Score:**

The paper has limited novelty and the method/theory/experiments are questionable.

**Justification For Why Not Lower Score:**

N/A

**Metareview: Summary, Strengths And Weaknesses:**

In this paper, the authors proposed a method aiming to solve the problem of preserving privacy of sharing the graph structure in training GNNs. They proposed adding noice by perturbing the edges of each node. The writing is not clear, the novelty is limited, and the correctness is questionable.